# A Retrospective Study to Compare the Incidence of Hyponatremia after Administration between Linezolid and Tedizolid

**DOI:** 10.3390/antibiotics12020345

**Published:** 2023-02-07

**Authors:** Yuichi Shibata, Mao Hagihara, Nobuhiro Asai, Arufumi Shiota, Jun Hirai, Nobuaki Mori, Hiroshige Mikamo

**Affiliations:** 1Department of Pharmacy, Aichi Medical University Hospital, Yazakokarimata, Nagakute 480-1195, Aichi, Japan; 2Department of Clinical Infectious Diseases, Aichi Medical University, Yazakokarimata, Nagakute 480-1195, Aichi, Japan

**Keywords:** Linezolid, Tedizolid, hyponatremia, albumin

## Abstract

Linezolid (LZD) and Tedizolid (TZD) are oxazolidinone antibiotic for meticillin-resistant *Staphylococcus aureus* (MRSA). Severe hyponatremia after LZD administration have been reported. Severe hyponatremia cause seizures, unconsciousness, and even death. Therefore, we conducted a study to assess the change of serum sodium level after LZD and TZD therapy. We enrolled 67 patients treated with LZD and 28 treated with TZD. We monitored the serum sodium level from the administration to 14 days after administration of oxazolidinone drug. Hyponatremia was defined a sodiuln level ≤134 mmol/L after the initiation of oxazolidinone drug. The frequency of hyponatremia in the LZD group was significantly higher than that in the TZD group (39.7% vs. 11.1%, *p* < 0.05). The rate of patients administered by injection was significantly higher than in the LZD group than in the TZD group (52.9% vs. 14.8%, *p* < 0.01). Multiple logistic regression analyses identified the albumin level before the oxazolidinone drug therapy as the independent variables associated with the development of hyponatremia. We revealed that TZD is safer than LZD in terms of hyponatremia. Therefore, cases that LZD is administered by injection should be used more carefully with hyponatremia in patients with low albumin level.

## 1. Introduction

Linezolid (LZD) is a first oxazolidinone antibiotic for meticillin-resistant *Staphylococcus aureus* (MRSA). Owing to high tissue penetration and almost 100% bioavailability, oral drugs can be expected to have the equivalent antibacterial activity as injection drugs. [1,2]. Tedizolid (TZD) is next oxazolidinone antibiotic approved after LZD. TZD has good tissue transferability and is used for skin, soft tissue, and osteoarticular infections [3,4]. In such conditions, long-term use may be necessary, which might lead to an increased risk of adverse events. Thrombocytopenia is a major side effect with LZD. Though in previous study reported that rates of thrombocytopenia were found to be lower with TZD than with LZD [5]. In addition to thrombocytopenia, the side effect of hyponatremia with LZD has also been reported [6,7]. Hyponatremia is the major electrolyte abnormality seen in the clinical setting. Severe hyponatremia is associated with increased mortality and morbidity and longer hospital stays [8,9]. Tedizolid is a drug that is expected to have improved safety compared to linezolid, especially with myelotoxicity and drug interactions [10,11,12]. However, Studies regarding the adverse effects of TZD in Japanese patients are scarce [13]. Comparing the incidence of hyponatremia between LZD and TZD is considered important to ensure patient safety in treating MRSA. Therefore, we conducted a comparative retrospective study to assess hyponatremia associated with LZD and TZD.

## 2. Methods

### 2.1. Patients

A total of 95 patients received LZD or TZD treatment at the Aichi Medical University Hospital between January 2017 and August 2022 were included in this study. Patients over the age of 20 years and received six or more doses of oxazolidinone drug. Patients undergoing dialysis and cared in the intensive care unit or emergency center for the duration of oxazolidinone treatment were excluded. Patients whose serum sodium levels were not measured at least twice or were less than l34 mmol/L before the initial oxazolidinone drug were also excluded. Inclusion and exclusion criteria were set by modifying previous reports [14]. We collected clinical data of patients, including age, sex, weight, formulation, concomitant medications, infection type, microbiological data and laboratory data. Concomitant drugs that may influence sodium levels were defined by previous reports [15,16,17]. The level of serum sodium was monitored from the administration to 14 days after administration of oxazolidinone therapy. We did not take any special measures against potential sources of bias in this retrospective study.

### 2.2. Evaluation of Hyponatremia

Hyponatremia was defined as a sodium level ≤134 mmol/L after the initiation of oxazolidinone therapy. The criteria of hyponatremia were also applied in the present study [14].

### 2.3. Analysis of Risk Factors Associated with Hyponatremia

The risk factors related to hyponatremia were analyzed by multiple logistic regression. We extracted the factors by stepwise selection and univariate analysis.

### 2.4. Statistical Analysis

No statistical sample size calculations were conducted. Categorical data were analyzed by Fisher test for categorical data, while continuous data were analyzed by Mann–Whitney U test. We used JMP, version 10.0 (SAS, Tokyo, Japan) for statistical analysis software. Statistical significance was set at *p* < 0.05. The relationship between serum albumin levels before oxazolidinone therapy and hyponatremia was analyzed using receiver operating characteristic (ROC) curve.

## 3. Results

### 3.1. Patients

Figure 1 shows the flowchart used in the study. We enrolled 207 patients in this study and excluded 112 since underwent hemodialysis, cared in the intensive care unit or emergency center and whose serum sodium levels were not measured at least twice or were less than l34 mmol/L before the initial oxazolidinone drug.

The patients treated with LZD and TZD was 68 and 27 were included in our study (Table 1). The duration in the TZD group were longer than those in the LZD group (9.0 days [6.0–14.0 days] vs. 15.0 days [11.0–35.0 days], *p* < 0.01). Regarding formulation, the rate of patients administered by injection was significantly higher in the LZD group than in the TZD group (52.9% vs. 14.8%, *p* < 0.01). On the other hand, the rate of patients administered by oral was significantly higher in the TZD group than in the LZD group (39.7% vs. 77.8%, *p* < 0.01). Regarding laboratory data, albumin value of before oxazolidinone treatment was significantly lower in the LZD group than in the TZD group (2.5 g/dL [2.1–3.1 g/dL] vs. 2.8 [2.5–3.4 g/dL], *p* < 0.05).

The infection types are presented in Table 2. The major infection types in the LZD group were respiratory infection and bacteremia (30.9% and 19.1%). On the other hand, the highest proportion in the TZD group were skin infection and wound infection (29.8% and 44.4%). There was significantly difference in the infection type between the LZD and TZD groups.

The proportion of patients based on the combination antimicrobial with oxazolidinone treatment is presented in Table 3. The proportion of patients used fluoroquinolones was significantly higher in the TZD group than in the LZD group (2.9% vs. 18.5%, *p* < 0.05).

The highest proportion of patients had Non-Steroidal Anti-Inflammatory Drugs (NSAIDs) in both of the groups (23.5% and 44.4%) (Table 4). There was no significant difference in the use condition between both groups.

### 3.2. Hyponatremia

Clinical data with hyponatremia after oxazolidinone treatment are presented in Table 5. The frequency of hyponatremia in LZD group was significantly higher than that in the TZD group (39.7% vs. 11.1%, *p* < 0.05). The period of hyponatremia was mostly during administration in both cases (81.5% vs. 66.7%). Furthermore, the time to reach sodium nadir in the LZD group had a tendency to be shorter than that in the TZD group (5.0 days [3.0–7.0 days] vs. 13.0 days [3.5–16.5] days, *p* < 0.05). Hence, reduction rate of Na was significantly higher in the LZD group than in the TZD group (2.2% [0.7–4.3%] vs. 0.7% [0–2.1%], *p* < 0.05).

The result of multiple logistic regression analysis is presented in the Table 6. We extracted the factors by stepwise selection and univariate analysis. We identified albumin levels before the initial oxazolidinone drug treatment and the LZD use of as the independent variables associated with the development of hyponatremia. The odds ratios (95% confidence interval [CI]) for albumin levels and LZD use were 0.33 (CI, 0.16–0.71) (*p* < 0.01) and 4.34 (CI, 1.12–16.80) (*p* < 0.05), respectively. Therefore, we analyzed the incidence of hyponatremia by renal function and albumin level (Table 6)**.**

To evaluate the contribution of patient’s renal function on the occurrence of hyponatremia due to oxazolidinone therapy, we divided patients into two groups depending on their estimated glomerular filtration rate (eGFR) at 60 mL/min/1.73 m^2^ (Table 7). As a result, the patients with high renal function (eGFR ≥ 60 mL/min/1.73 m^2^) used LZD had hyponatremia significantly higher than patients used TZD (44.0% vs. 10.5%, *p* < 0.05). Regarding with albumin, we divided patients into two groups depending on their albumin at 2.6 g/dL. As a result, the patients with high albumin level (albumin ≥ 2.6 g/dL) who used LZD had hyponatremia significantly higher than patients who used TZD (22.6% vs. 0%, *p* < 0.05). In patients with low albumin levels (albumin < 2.6 g/dL), hyponatremia was happened in approximately 50% in both groups (54.1% and 42.9%).

The relationship with albumin and hyponatremia was shown in the Figure 2 and Figure 3. The albumin cut-off value was set at 2.6 g/dL in the LZD group, which had a sensitivity of 56.1%, a specificity of 81.5%. On the other hand, the albumin cut-off value was set at 2.5 in the TZD group, which had a sensitivity of 82.6%, a specificity of 100.0%.

## 4. Discussion

In our study, the incidents of hyponatremia and reduction rate of sodium level were higher in the LZD group than in the TZD. Hence, the time to reach sodium nadir in the LZD group had a tendency to be shorter than that in the TZD group. It was considered that the results may influenced by the serum albumin level before initiation of oxazolidinone drug treatment and the rate of patients used injectable drug. Albumin is among the most important proteins and plays a significant role in maintenance of colloid osmotic pressure [18,19,20,21,22]. It is reported that decreasing albumin level induce low blood volume and stimulated renin-angiotensin-aldosterone system and arginine-vasopressin system [23]. Therefore, the water retention and hyponatremia occurred by active renin-angiotensin-aldosterone system and arginine-vasopressin system [23,24,25]. Based on these reports, it was considered that hyponatremia was more likely to occur in the LZD group with low albumin levels than TZD group. In our results, albumin level of pre-administration of oxazolidinone antibiotics was associated with hyponatremia (Figure 2 and Figure 3). These results revealed that the albumin level of initial oxazolidinone drug treatment could possibly estimate the incidence of hyponatremia. Ongoing inflammation derived from severe infection may cause a reduction in synthesis and an increase in catabolism of albumin [26,27]. Furthermore, in patients with low albumin levels (albumin < 2.6 g/dL), approximately 50% hyponatremia was observed in both groups (Table 5), and monitoring sodium level is required. Regarding formulation, the rate of patients administrated by injection was significantly higher in the LZD group than in the TZD group (52.9% vs. 14.8%, *p* < 0.01). Injection of LZD contain 300 mL of fluid without sodium and recommended twice daily, resulting in a total of 600 mL fluid without sodium being loaded. In contrast, the injection of TZD is recommended to be diluted in 250 mL of saline with sodium and administered once daily. It was considered that hyponatremia was induced by the dilution of body fluids due to fluid without sodium loading by LZD and the pharmacological action of oxazolidinone drug. Several considerations should be made when interpreting our results. First, this was a retrospective study with small sample size. Thus, further investigation in a large population is needed to support the results of safety. Second, there was large difference in the population between LZD and TZD group. Indication disease for TZD is considerably less than LZD, limited to skin infections and postoperative infections. Due to the difference of indication disease for both drugs, there was significantly difference in the diseases, concomitant antibiotics and duration of treatment between LZD and TZD group. Furthermore, serum albumin levels before administration in the LZD group was significantly lower than in the TZD group. Our result revealed the relationship between pre-albumin levels and hyponatremia. The difference of serum albumin levels between the two groups may have caused our results to become biased. Therefore, it could not be denied that the results of this study have low external validity. Finally, we did not measure the blood oxazolidinone antibiotics concentrations. Previous studies have shown that the renal excretion rate of linezolid is 30% and its accumulation in the body causes adverse events in patients with decreased renal function [28]. Moderate renal disease (a creatinine clearance (CLCR) 30–50 mL/min) was the risk factor of thrombocytopenia derived from LZD [29,30]. The half-life of TZD in the hemodialysis (eGFR < 15 mL/min/1.73 m^2^) patients and non-hemodialysis (eGFR < 30 mL/min/1.73 m^2^) patients were 11.4 h and 12.9 h. In contrast with LZD, there was no difference in pharmacokinetics due to renal clearance [31]. Previous reports provided that a pharmacokinetic explanation for the mechanism of the adverse event that renal dysfunction increased linezolid trough concentration and AUC and that higher drug exposure induced thrombocytopenia [32,33,34]. However, the patients with high renal function (eGFR ≥ 60 mL/min/1.73 m^2^) who used LZD had hyponatremia significantly higher than patients used TZD (44.0% vs. 10.5%, *p* < 0.05) in our study. Therefore, it could not be denied that hyponatremia was not derived from oxazolidinone antibiotics.

## 5. Conclusions

In conclusion, we revealed that TZD is safer than LZD in terms of hyponatremia. However, hyponatremia occurred more than 10% in both groups, and sodium level monitoring was required after oxazolidinone antibiotics treatment. Therefore, cases where LZD is administered by injection should be used more carefully in patients with albumin level less than 2.6 g/dL and sodium levels should be monitored more carefully.

## Figures and Tables

**Figure 1 antibiotics-12-00345-f001:**
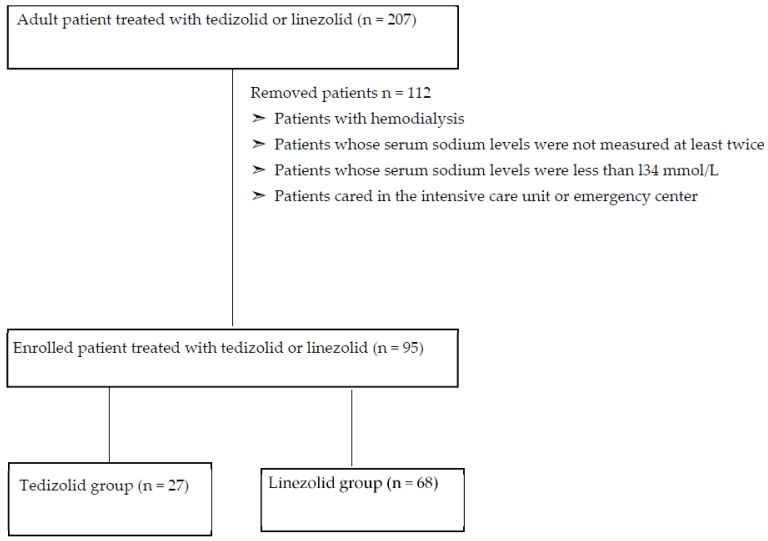
Shows the flow-chart of this study.

**Figure 2 antibiotics-12-00345-f002:**
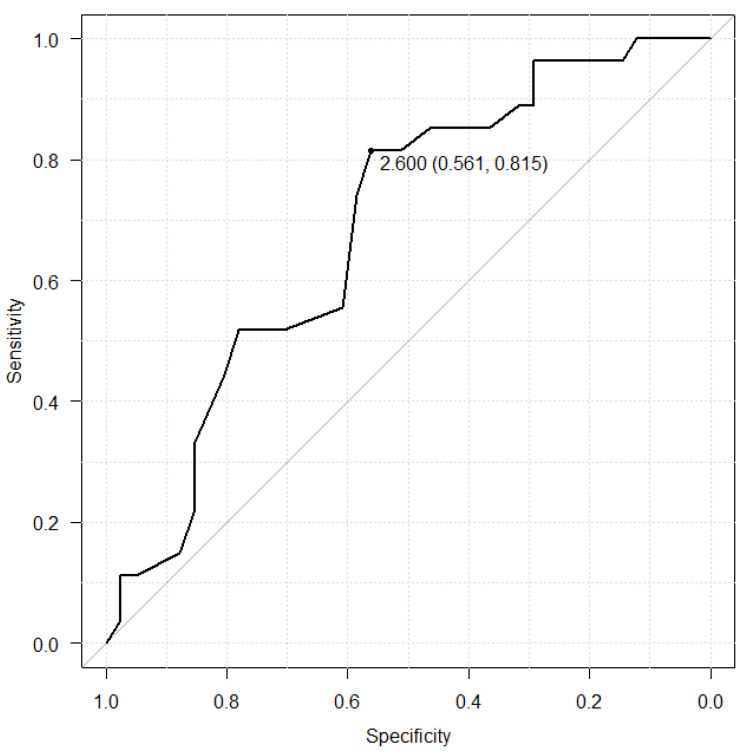
Receiver operating characteristic (ROC) curve between serum albumin level before LZD treatment and hyponatremia.

**Figure 3 antibiotics-12-00345-f003:**
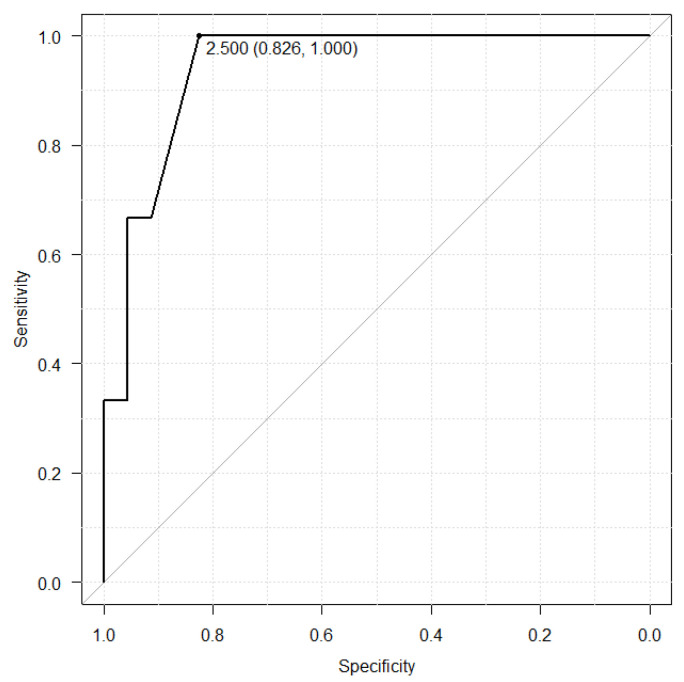
Receiver operating characteristic (ROC) curve between serum albumin level before TZD treatment and hyponatremia.

**Table 1 antibiotics-12-00345-t001:** Clinical characteristics of patients.

	LZD (*n* = 68)	TZD (*n* = 27)	*p* Value
Duration (days)	9.0 (6.0–14.0)	15.0 (11.0–35.0)	<0.01 ^a^
Age (years)	74.5 (51.8–83.0)	67.0 (52.0–78.5)	0.29 ^a^
BMI (kg/m^2^)	21.5 (18.6–23.3)	21.4 (19.1–25.4)	0.54 ^a^
Sex (male/female)	38/30	20/7	1.0 ^b^
Formulation			
Injection (%)	52.9 (36/68)	14.8 (4/27)	<0.01 ^c^
Oral (%)	39.7 (27/68)	77.8 (21/27)	<0.01 ^c^
Both injection and oral (%)	7.4 (5/68)	7.4 (2/27)	1.0 ^d^
Bacterial species			
MRSA (%)	45.6 (31/68)	44.4 (12/27)	0.92 ^c^
MRCNS (%)	14.7 (10/68)	14.8 (4/27)	1.0 ^d^
Others (%)	39.7 (27/68)	40.8 (11/27)	0.93 ^c^
WBC (×10^3^/μL)	8.9 (6.9–12.3)	7.7 (5.7–10.3)	0.05 ^a^
Albumin (g/dL)	2.5 (2.1–3.1)	2.8 (2.5–3.4)	<0.05 ^a^
eGFR (mL/min/1.73 m^2^)	75.0 (42.8–110.8)	74.3 (47.3–146.1)	0.48 ^a^
Na (mmol/L)	139.0 (136.0–141.0)	139.0 (136.5–141.5)	0.78 ^a^
K (mmol/L)	3.9 (3.5–4.3)	4.1 (3.8–4.4)	0.28 ^a^
Cl (mmol/L)	103.0 (100.0–105.0)	102.0 (100.0–106.0)	0.80 ^a^
AST (U/L)	25.0 (18.8–37.8)	25.0 (18.5–52.5)	0.46 ^a^
ALT (U/L)	20.0 (12.0–42.0)	29.0 (15.0–71.0)	0.12 ^a^
CRP (mg/dL)	7.1 (3.2–12.4)	3.3 (1.0–11.0)	0.09 ^a^

BMI: body mass index. MRSA: Methicillin-resistant Staphylococcus aureus. MRCNS: Methicillin-resistant coagulase negative Staphylococci. WBC: white blood cell. eGFR: estimated glomerular filtration rate. AST: aspartate aminotransferase. ALT: alanine aminotransferase. CRP: C-reactive protein. ^a^ Mann–Whitney U test for continuous data (median (IQR)); ^b^ Chi square test (yates correction) for categorical data; ^c^ Chi square test for categorical data; ^d^ Fisher’s test for categorical data.

**Table 2 antibiotics-12-00345-t002:** Proportion of patients based on the infection type.

Infection Type	LZD *n* (%)	TZD *n* (%)	*p* Value
Respiratory infection	21 (30.9)	0 (0)	<0.01 ^a^
Skin tissue infection	11 (16.2)	8 (29.6)	0.23 ^a^
Bacteremia	13 (19.1)	1 (3.7)	<0.01 ^b^
Wound infection	6 (8.8)	12 (44.4)	<0.01 ^a^
Meningitis	4 (5.9)	0 (0)	0.56 ^b^
Osteomyelitis	7 (10.3)	3 (11.1)	1.0 ^b^
Gastrointestinal infection	3 (4.4)	0 (0)	0.56 ^b^
Arthritis	2 (2.9)	2 (7.4)	0.32 ^b^
Urinary infection	6 (8.8)	0 (0)	0.18 ^b^
Others	8 (11.8)	2 (7.4)	0.72 ^b^

^a^ Chi square test for categorical data; ^b^ Fisher’s test for categorical data.

**Table 3 antibiotics-12-00345-t003:** Combination antimicrobial with oxazolidinone treatment.

	LZD *n* (%)	TZD *n* (%)	*p* Value
Penicillin	19 (27.9)	9 (33.3)	0.79 ^a^
Cephalosporin	10 (14.7)	0 (0)	0.06 ^b^
Carbapenem	15 (22.1)	2 (7.4)	0.17 ^c^
Lincomycin	4 (5.9)	0 (0)	0.58 ^b^
Fluoroquinolone	2 (2.9)	5 (18.5)	<0.05 ^b^
Metronidazole	4 (5.9)	1 (3.7)	1.0 ^b^
Tetracycline	2 (2.9)	1 (3.7)	1.0 ^b^
Sulfamethoxazole/ Trimethoprim	2 (1.4)	1 (3.7)	1.0 ^b^
Antifugal	1 (1.5)	1 (3.7)	0.49 ^b^
Others	7 (10.3)	2 (7.4)	1.0 ^b^
No drugs	20 (14)	13 (48.1)	0.14 ^c^

^a^ Chi square test for categorical data; ^b^ Fisher’s test for categorical data; ^c^ Chi square test (yates correction) for categorical data.

**Table 4 antibiotics-12-00345-t004:** Concomitant drugs that may influence serum sodium levels.

	LZD *n* (%)	TZD *n* (%)	*p* Value
ARB	11 (16.2)	7 (25.9)	0.43 ^a^
ACE inhibitor	3 (4.4)	0 (0)	0.56 ^b^
Potassium sparing diuretic	4 (5.9)	3 (11.1)	0.40 ^b^
NSAIDs	16 (23.5)	12 (44.4)	0.08 ^a^
SGLT2 inhibitor	1 (1.5)	1 (3.7)	0.49 ^b^
SSRI	1 (1.5)	0 (0)	1.0 ^b^
Opioid	2 (2.9)	0 (0)	0.49 ^b^
Carbamazepine	1 (1.5)	0 (0)	1.0 ^b^
Steroids	11 (16.2)	5 (18.5)	0.98 ^a^
Thiazide diuretic	0 (0)	1 (3.7)	0.28 ^b^
Sodium valproate	2 (2.9)	0 (0)	0.49 ^b^
Loop diuretic	8 (11.8)	3 (11.1)	1.0 ^b^
Tolvaptan	2 (2.9)	1 (3.7)	1.0 ^b^
No drugs	27 (39.7)	6 (22.2)	0.17 ^a^

ARB: Angiotensin II Receptor Blocker. ACE inhibitor: Angiotensin Converting Enzyme inhibitor. NSAIDs: Non-Steroidal Anti-Inflammatory Drugs. SGLT2 inhibitor: Sodium-glucose cotransporter 2 inhibitors. SSRI: Selective Serotonin Reuptake Inhibitor. ^a^ Chi square test (yates correction) for categorical data; ^b^ Fisher’s test for categorical data.

**Table 5 antibiotics-12-00345-t005:** Clinical data with hyponatremia after oxazolidinone treatment.

	LZD *n* (%)	TZD *n* (%)	*p* Value
Hyponatremia	27/68 (39.7)	3/27 (11.1)	<0.05 ^a^
The time to hyponatremia (Day)	4.0 (2.5–6.0)	4.0 (3.5–9.0)	0.60 ^b^
The period of hyponatremia			0.50 ^c^
During administration (%)	22/27 (81.5)	2/3 (66.7)	
After administration (%)	5/27 (18.5)	1/3 (33.3)	
Minimum value (mmol/L)	131.0 (130.0–132.0)	138.0 (136.0–139.0)	0.07 ^b^
Nadir (mmol/L)	5.0 (3.0–7.0)	13.0 (3.5–16.5)	<0.05 ^b^
Reduction rate (%)	2.2 (0.7–4.3)	0.7 (0–2.1)	<0.05 ^b^

^a^ Chi square test (yates correction) for categorical data; ^b^ Mann–Whitney U test for continuous data (median (IQR)); ^c^ Fisher’s test for categorical data.

**Table 6 antibiotics-12-00345-t006:** Multivariate analysis of demographic and clinical data in the no-hyponatremia and hyponatremia groups.

Covariate Multivariate Analysis	Odds Ratio	95% CI	*p* Value
Albumin	0.33	0.16–0.71	<0.01
LZD administration	4.34	1.12–16.80	<0.05

95% CI: 95% confidence interval.

**Table 7 antibiotics-12-00345-t007:** Incidence of hyponatremia compared by renal function and albumin.

	LZD *n* (%)	TZD *n* (%)	*p* Value
Renal function			
eGFR ≥ 60 mL/min/1.73 m^2^ (LZD: *n* = 50, TZD: *n* = 19)	22/50 (44.0)	2/19 (10.5)	<0.05 ^a^
eGFR < 60 mL/min/1.73 m^2^ (LZD: *n* = 18, TZD: *n* = 7)	5/18 (27.8)	1/7 (14.3)	0.64 ^b^
Albumin			
Albumin ≥ 2.6 g/dL (LZD: *n* = 31, TZD: *n* = 19)	7/31 (22.6)	0/19 (0)	<0.05 ^b^
Albumin < 2.6 g/dL (LZD: *n* = 37, TZD: *n* = 7)	20/37 (54.1)	3/7 (42.9)	0.69 ^b^

^a^ Chi square test (yates correction) for categorical data; ^b^ Fisher’s test for categorical data.

## Data Availability

The datasets analyzed in this study are available and can be obtained upon reasonable request.

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
