# Peer review of "A Retrospective Study to Compare the Incidence of Hyponatremia after Administration between Linezolid and Tedizolid"

_antibiotics, 2023, doi:10.3390/antibiotics12020345_

Round 1

Reviewer 1 Report

Dear authors,

This study presents the results of a retrospective study about the incidence of hyponatremia in LZD compared to TZD population. This is a safety issue for a antibiotic choice in patients with hyponatremia and/or hypoalbuminemia.

However, I do have several questions, comments, and suggestions.

My major concern is about methods presented after results and discussion, this is unusal.

The methods should follow STROBE guidelines. The choice of sample size could be explained.

Difference of patient’s characteristics at baseline (more hypoalbuminemia in LZD group) should be presented as a major bias in results interpretation, as hypoalbuminemia is an identified risk factor of hyponatremia.

Minor concerns: Ref [7], in the final results of the 2 RCT studies, there are more diarrhea and vomiting in LZD compared to TZD group, that could be discussed as a mechanism of hyponatremia?

I would be very interested to read your corrected version,

Best regards

Author Response

Thank you for pointing out.
I have attached the revised manuscript, so please check it.

Reviewer 2 Report

The methodology seems correct and sounds good. Some corrections could improve the quality of the work.

1) Please move the methods paragraph after the introduction and before the results.

2) In methods, a non-parametric test was performed to analyze continuous data but you do not indicate how the data are distributed. Please add a normality test of data distribution.

3)in the bibliography the 5 and 6 references must be correct according to the author's guide

Author Response

(The authors gave the same response as above.)

Round 2

Reviewer 1 Report

I read carefully this new version. What about the methodology of reference STROBE Guidelines? are there followed?

minor concerns : l48: this is not a sentence

l76: this sentence sounds odd.

Author Response

 I have a question about the reviewer's comments.

・l48: this is not a sentenceAs a result, the patients with high renal function (eGFR ≥ 60mL/min/1.73m2) used LZD were happened hyponatremia significantly higher than patients used TZD (44.0 % vs. 10.5 %, p < 0.05).

・l76: this sentence sounds odd.

 It is reported that decreasing albumin level induce low blood volume and stimulated renin-angiotensin-aldosterone system and arginine-vasopressin system.I don't think either of these sentences are so inconsistent.Please let me know if the point you pointed out is different.

Reviewer 2 Report

All suggestions have been fixed

Author Response

thank you for your review

Round 3

Reviewer 1 Report

All suggestions have been fixed